# Etiology of Breast Implant-Associated Anaplastic Large Cell Lymphoma (BIA-ALCL): Current Directions in Research

**DOI:** 10.3390/cancers12123861

**Published:** 2020-12-21

**Authors:** Anand K. Deva, Suzanne D. Turner, Marshall E. Kadin, Mark R. Magnusson, H. Miles Prince, Roberto N. Miranda, Giorgio G. Inghirami, William P. Adams

**Affiliations:** 1Department of Plastic and Reconstructive Surgery, Macquarie University and the Integrated Specialist Healthcare Education and Research Foundation, Sydney, NSW 2109, Australia; 2Division of Cellular and Molecular Pathology, Department of Pathology, University of Cambridge, Cambridge CB2 1TN, UK; sdt36@cam.ac.uk; 3Senior Researcher, Šárka Pospíšilová Research Group, CEITEC, Masaryk University, 601 77 Brno, Czech Republic; 4Department of Dermatology, Roger Williams Medical Center, Providence, RI 02908, USA; mkadin@me.com; 5Boston University School of Medicine, Boston, MA 02908, USA; 6Department of Plastic Surgery, School of Medicine, Griffith University, Southport, QLD 4222, Australia; mark@drmagnusson.com.au; 7Epworth Healthcare, East Melbourne, Richmond, VIC 3121, Australia; Miles.Prince@petermac.org; 8Sir Peter MacCallum Department of Oncology, University of Melbourne, Parkville, VIC 3000, Australia; 9Department of Hematopathology, The University of Texas MD Anderson Cancer Center, Houston, TX 77030, USA; roberto.miranda@mdanderson.org; 10Department of Pathology, Weill Cornell Medicine, New York, NY 10065, USA; ggi9001@med.cornell.edu; 11Department of Plastic Surgery, University of Texas Southwestern Medical Center, Dallas, TX 75390, USA; wpajrmd@dr-adams.com

**Keywords:** antigens, bacterial, breast implants, lymphoma, T-cells

## Abstract

**Simple Summary:**

The first report of breast implant-associated anaplastic large cell lymphoma (BIA-ALCL) was in 1997. Although BIA-ALCL develops around breast implants, it is considered a cancer of the immune system and not a cancer of the breast ducts or lobules. Nearly all confirmed cases to date have been associated with textured surface (versus smooth surface) breast implants. As physicians have become more aware of BIA-ALCL, so has the number of reported cases, although the number of cases remains low. In most instances, patients have an excellent prognosis following removal of the breast implant and its surrounding fibrous capsule. Many theories on factors that trigger the development of BIA-ALCL, such as the presence of bacteria, have been proposed. However, the sequence(s) of events that follow the initial triggering event(s) have not been fully determined. This article summarizes the current scientific knowledge on the development of BIA-ALCL.

**Abstract:**

Breast implant-associated anaplastic large cell lymphoma (BIA-ALCL) is a CD30-positive, anaplastic lymphoma kinase-negative T-cell lymphoma. Where implant history is known, all confirmed cases to date have occurred in patients with exposure to textured implants. There is a spectrum of disease presentation, with the most common occurring as a seroma with an indolent course. A less common presentation occurs as locally advanced or, rarely, as metastatic disease. Here we review the immunological characteristics of BIA-ALCL and potential triggers leading to its development. BIA-ALCL occurs in an inflammatory microenvironment with significant lymphocyte and plasma cell infiltration and a prominent Th1/Th17 phenotype in advanced disease. Genetic lesions affecting the JAK/STAT signaling pathway are commonly present. Proposed triggers for the development of malignancy include mechanical friction, silicone implant shell particulates, silicone leachables, and bacteria. Of these, the bacterial hypothesis has received significant attention, supported by a plausible biologic model. In this model, bacteria form an adherent biofilm in the favorable environment of the textured implant surface, producing a bacterial load that elicits a chronic inflammatory response. Bacterial antigens, primarily of Gram-negative origin, may trigger innate immunity and induce T-cell proliferation with subsequent malignant transformation in genetically susceptible individuals. Although much remains to be elucidated regarding the multifactorial origins of BIA-ALCL, future research should focus on prevention and treatment strategies, recognizing susceptible populations, and whether decreasing the risk of BIA-ALCL is possible.

## 1. Introduction

Breast implant‒associated anaplastic large cell lymphoma (BIA-ALCL), first reported in 1997 [1], is a CD30-positive, anaplastic lymphoma kinase (ALK) ‒negative T-cell lymphoma [2,3]. Where implant history is known, BIA-ALCL has been reported only in patients with textured breast implants or patients with a history of a textured device [3,4,5]. The United States Food and Drug Administration has included cases of BIA-ALCL in patients with smooth implants in medical device reports of BIA-ALCL; however, patient history may be incomplete, and implant replacements over time may not have been recorded per methodologic approach to include reportable cases [6]. As awareness of BIA-ALCL as a distinct clinical condition has grown, so too has the number of reported cases, although the number of cases remains low globally [3,7]. In the United States, an incidence rate of 2.03 per million person-years from 1996 to 2015 has been reported, corresponding to a lifetime risk of 1 per 30,000 women with textured implants [4]. This study was based on 100 cases in the United States and, currently, more unique cases have been reported [6], which may affect the risk calculation. In a consecutive series of patients who received textured implants in a large cancer center in the United States, there was an estimated incidence of 1 per 355 women, perhaps suggesting a higher sensitivity to developing BIA-ALCL among patients with a history of breast cancer [8,9]. Alternatively, this may represent a significant case cluster, which would suggest some alteration or break in surgical technique, thus potentiating cases. The incidence of BIA-ALCL with other plastic surgeons at that same cancer center would be an important comparator. In the Netherlands, the cumulative risk was 29 per million at 50 years and 82 per million at 70 years, or an overall risk of 1 per 6920, in women receiving implants from 1965 to 2015 [10]. Varying incidence rates between countries and regions may reflect differences in genetic backgrounds, levels of awareness, the completeness of screening and assessment of surgical specimens, reporting options, breast microbiome, and the availability of textured surface implants, particularly macrotextured surfaces such as polyurethane and Biocell (Allergan plc, Dublin, Ireland). The risk of developing BIA-ALCL does not appear to be dependent on the age of the patient [11]. However, in a systematic review, breast reconstruction patients were found to be older at the onset of BIA-ALCL than those undergoing augmentation, which was attributed to the younger age at which breast augmentation is generally performed [12]. National registries have been established to support the reporting of BIA-ALCL, and a recent initiative has been proposed to integrate the separate registries [13].

Although BIA-ALCL was provisionally classified as a distinct disease by the World Health Organization in 2016 [14], it is now considered a definitive entity [15,16]. Differences in outcomes between patients with early-stage disease presenting with seroma only and those with a mass at presentation may suggest two distinct clinicopathological entities [17], but it is also likely they represent the spectrum of tumor progression [18]. An alternative concept suggested by some scientists is that BIA-ALCL may be better classified as a lymphoproliferative disease in view of the indolent nature of early-stage disease and a more aggressive path for advanced disease [19,20,21]; however, confirmation will require further analysis of clinical cases including clonality of T-cell receptor (TCR) gene rearrangements, chromosomal abnormalities, single-nucleotide variants, host genotyping, and cytokine expression, especially by comparison to and with thorough examination of benign inflammatory seromas [21].

Despite the evolving science, the prognosis for BIA-ALCL remains excellent, and the risk of death from BIA-ALCL is very low compared to other malignancies [22]. For the majority of patients with BIA-ALCL restricted to the fibrous capsule surrounding the breast implant (i.e., effusion-limited), optimal management consists of timely diagnosis and surgical excision of implants and capsule with negative margins [23]. Complete surgical excision significantly improves event-free and overall survival relative to other treatment strategies [24].

Our current understanding of BIA-ALCL pathophysiology is incomplete and fragmentary, and there is a need for evidence-based conclusions. Multiple BIA-ALCL etiologies have been proposed with varying degrees of support. This paper provides a summary of the immunological characteristics of BIA-ALCL and the evidence behind each proposed etiology, identifies data gaps, and proposes directions for future research aimed at elucidating the origins of BIA-ALCL.

## 2. Methods

On 26 April 2018, an international, multidisciplinary meeting that included experts in the fields of plastic surgery, oncology, pathology, dermatology, and hematology was convened in New York City. Its objective was to garner opinions regarding the etiology of BIA-ALCL based on published and ongoing research, as well as perspectives on clinical experience. Data examined were based on participants’ analyses of the published literature, as well as their own ongoing research, and opinions expressed were their own rather than representative of any institution or group. A summary of the meeting, representing the panel’s expert opinion, was prepared for use in a publication aimed at briefing plastic surgeons on the current state of knowledge regarding BIA-ALCL immunology and etiology. The manuscript has evolved since the meeting via email discussion and revision to reflect the most up-to-date literature.

## 3. BIA-ALCL Immunological Characteristics

The development of BIA-ALCL appears to be mediated by a combination of innate and adaptive immunological processes. Innate immunity is a process by which cells such as macrophages and neutrophils non-specifically clear pathogens without the possibility of immunological memory. Three groups of innate lymphoid cells (ILCs) have also been shown to play a role in innate immunity [25]. In contrast, adaptive immunity is primarily a B- and T-lymphocyte‒mediated, antigen-specific response capable of immunological memory [26,27].

While the role of innate immunity in the pathogenesis of BIA-ALCL is not clearly defined, it cannot be excluded from playing an important role. The initial innate immune response may eventually elicit an adaptive immune response, resulting in recruitment of T-cells to the site of inflammation. However, whether the subsequent lymphomagenic process occurs in an antigen-dependent manner is unknown. BIA-ALCL cell lines and tissue from most patients with BIA-ALCL lack expression of cell surface TCRs, suggesting that either TCRs are down-regulated during lymphomagenesis or that they are never expressed to begin with, despite the incipient tumor cells having had the capacity to do so (as evidenced by the presence of monoclonal TCR gene rearrangements) [18,28,29]. In support of the latter, ILCs have been described that lack a surface or membrane TCR yet share many other characteristics with antigen-specific T-cells [25,30]. For example, the group 3 subtype of ILCs share many similarities with T-helper (Th)17 cells but in contrast lack a TCR [25,30]. Therefore, in the case of BIA-ALCL, the cytokines and chemokines released by innate immune cells may activate the adaptive immune response [30]. The T-cells subsequently recruited to the site of inflammation would then respond in an antigen-dependent manner through the expressed TCR, which is subsequently down-regulated. Alternatively, the T-cells may act in an antigen-independent manner due to the lack of expression of a cell-surface TCR, similar to the response of ILCs to an inflammatory milieu. Another explanation would be that the cells of origin of BIA-ALCL are ILCs; however, this is unlikely because ILCs are unable to undergo clonal selection [30].

In adaptive immunity, CD4+ T-cells are activated in a process guided largely by the type of cytokines secreted by antigen-presenting cells. Activated CD4+ T-cells can differentiate into T helper 1 (Th1) CD4+ cells, which are primarily involved in cell-mediated antigen responses, T helper 2 (Th2) CD4+ cells, involved in B-cell responses to antigens and allergy, T helper 17 (Th17) CD4+ cells, which are induced by cytokines to enhance the inflammatory response, T-regulatory (Treg) CD4+ cells, which suppress immune response, and other subtypes [31]. A proinflammatory microenvironment with presumed chronic T-cell stimulation (as evidenced by CD30 expression—a cell surface receptor present on activated lymphoid cells) is a prominent feature of BIA-ALCL. Capsular tissue contains considerable amounts of cytokines (interleukin (IL)-1β and IL-6) promoting the differentiation and proliferation of Th17 cells, which maintain the inflammatory response [32]. Other inflammatory cells include eosinophils and mast cells with bound IgE, characteristic of allergic inflammation [17,33]. In addition, IL-13, the signature cytokine of allergic inflammation, has been identified in both BIA-ALCL cell cultures and in clinical samples [33,34]. Other cytokines have been identified in BIA-ALCL cell cultures, including IL-6, IL-9, and IL-10 [34].

Cytokine receptors are critical for the T-cell immune response [35]. Binding of interleukins to cytokine receptors induces intracellular signaling via the Janus kinase/signal transducer and activator of transcription (JAK/STAT) pathway, leading to changes in gene transcription [35]. Dysregulation of JAK/STAT signaling has been linked to cancers associated with inflammation [35,36,37,38,39]. In BIA-ALCL, STAT3 activation was detected in all 12 patient samples examined in one study, often due to the acquisition of a genetic defect within the pathway [17,40]. A single genetic lesion affecting JAK/STAT signaling in T-cells is unlikely to result in BIA-ALCL, and in vitro studies suggest that other factors, including IL-2 and IL-6 overexpression, dysregulation of survivin, and aberrantly low levels of the regulatory phosphatase Src homology region 2 domain-containing phosphatase-1, are necessary for BIA-ALCL development [29,41]. Genetic factors continue to be explored in the development of BIA-ALCL. Recent data have shown frequent mutations in epigenetic modifiers in patients with BIA-ALCL [40]. In one study involving 13 patients, the human leukocyte antigen allele A*26 was found significantly (*p* < 0.001) less often compared with the general population, which may reflect a genetic predisposition for the development of BIA-ALCL [42]. Larger studies would be needed to confirm this finding.

Among T-cell malignancies, the clinical course of BIA-ALCL is, in most cases, indolent, similar to primary cutaneous ALCL (pcALCL); both lymphomas consistently express CD30, similar to systemic ALCL (sALCL) [14,43]. However, BIA-ALCL is genetically distinct from these lymphomas and has been shown not to carry the chromosomal rearrangements observed in pcALCL and ALK-negative sALCL [44]. Elevated cytokine expression associated with the Th1 and Th17 phenotypes (interferon-γ and IL-17F, respectively) has been found in both BIA-ALCL and pcALCL cell lines [45], suggesting that discoveries in pcALCL may help guide research on the pathogenesis of BIA-ALCL.

Thus, the immunological processes necessary for the development of BIA-ALCL are evidently complex, with possible contributions from the innate immune system and dysregulation of multiple pathways in the adaptive immune system. In the remainder of this review, we discuss the etiology of BIA-ALCL.

## 4. Etiology of BIA-ALCL

### 4.1. Mechanical Friction

In patients with chronic inflammatory conditions, metal-on-metal joint replacement is associated with an increased risk of lymphoma [46,47]. However, cases of CD30-positive, ALK-negative ALCL associated with any type of prosthesis other than breast implants are extremely rare [48]. A 2013 review article identified a single such case, one that was associated with a stainless-steel orthopedic fixation plate [48,49]. Diffuse large B-cell lymphoma has been more frequently reported with various implanted devices [48]. Similar to BIA-ALCL, case reports of CD30-positive ALK-negative ALCL in patients with dental [50], gastric lap band [51], gluteal [52], and silicone-containing port device implants [53] have subsequently emerged [48]. While theories of lymphomagenesis by a proposed mechanism of mechanical friction have been suggested [54], evidence remains limited in light of the fact that the capsular-implant interface is typically “slick”.

### 4.2. Silicone Implant Shell Particulates

In the orthopedic literature, cases of silicone synovitis in patients with silicone elastomer implants used for joint reconstruction have been reported and attributed to an immunological reaction to particulate matter released from the implant [55,56]. A study of particles present around failed silicone orthopedic implants identified billions of particles less than 1 μm in size [57]. Failure of metal and polyethylene implants is linked to an innate immune system response in which activated macrophages become overwhelmed with particulate wear debris and subsequently release inflammatory factors that indirectly promote a T-cell response by lymphocyte chemotaxis and replication [54,58]. Such a mechanism—involving the capture of silicone-containing particles by macrophages followed by macrophage activation, cytokine production, and apoptosis—has been proposed for silicone-induced granuloma of the breast implant capsule [59]. Foreign-body reaction to silicone itself, its particles, or particles combined with autologous proteins is a possible trigger for the Th1/Th17 cell phenotype observed in peri-silicone implant capsular fibrosis [32,60]. The fact that greater silicone particle shedding is expected with textured implants compared with smooth implants [29] or is relative to the degree of texturing [61] is consistent with the particle trigger hypothesis. Data from research in Australia and New Zealand show that there is a high risk of BIA-ALCL with polyurethane-coated implants as well as silicone-coated implants. In addition, a cluster pattern of incidence has been observed that could be consistent with nosocomial contamination [62]. These data appear to be in conflict with the results of studies lending support to the silicone implant shell particulate theory.

### 4.3. Leachables

Conceivably, components of silicone breast implants could eventually leach across the implant shell and into surrounding tissue in soluble form, initiating an inflammatory response sufficient to trigger BIA-ALCL. Possible candidates include low molecular weight siloxanes and/or silicone gel, plasticizers, and platinum used as a catalyst in silicone polymerization [63]. Although one study found that silicone gel may activate B-cells, inducing neoplasms (i.e., characterized as plasmacytomas; however, no immunologic studies were performed to support the plasma cell differentiation) in genetically predisposed immune-compromised mice [64], evidence to support a role for siloxanes in human lymphomagenesis is lacking, and no correlation between BIA-ALCL incidence and silicone gel or saline-filled implants is evident. A review of studies on platinum leaching from breast implants and its possible biological effects concluded that “there are no clinical consequences of the platinum in silicone breast implants”, a view with which the Food and Drug Administration has concurred [65,66].

### 4.4. Bacteria/Biofilms

The link between infection and capsular contracture, one of the most common complications of breast implantation [67] and found in some women with BIA-ALCL [62], is well established [68,69,70]. Baker grade of capsular contracture has been shown to be significantly correlated with the presence of bacteria (*p* < 0.001, trend from Baker grade I (breast is soft and looks natural) through IV (breast has obvious severe contracture)) [68,71]. Of note, the human breast, rather than representing an aseptic environment, contains its own characteristic microbiome, which has been shown to be distinct from those of breast skin and buccal microbiomes [72,73]. This being the case, the bacterial source found in breasts with implants may be either endogenous or introduced during surgery [74]. Regardless of source, planktonic bacteria may attach to the implant surface in a self-produced matrix of extracellular polymeric substances known as a biofilm (Figure 1) [75]. The resulting environment provides protection for survival from antibiotics and impedes the host immune response [76,77]. The protective matrix may account for the chronic inflammation and long delay in development of BIA-ALCL. Surgical techniques that decrease the bacterial load around any surgical implantable device, including breast implants, have been described and are associated with a decrease in device-associated infection [78]. A prospective study found that the incidence of capsular contracture was lower than expected when betadine triple (povidone-iodine/cefazolin/gentamicin) and nonbetadine triple (bacitracin/cefazolin/gentamicin) antibiotic breast irrigation was used [79]. A 10-year prospective study of 17,656 patients with textured implants demonstrated that betadine pocket irrigation, but not non-betadine triple antibiotic pocket irrigation, was associated with decreased capsular contracture in the primary reconstruction cohort [80]. A meta-analysis found that povidone-iodine irrigation was significantly associated with a decrease in Baker class III/IV capsular contracture in patients undergoing aesthetic breast augmentation (2.7% vs. 8.9% for saline irrigation; *p* < 0.00001) [81]. Thirty studies have now supported the use of antimicrobial breast pocket irrigation with reduced rates of adverse events [82].

Several lines of evidence suggest that biofilm on the surface of breast implants provides the trigger for BIA-ALCL. In vitro and in vivo models showed higher bacterial loads, with significantly more bacteria attaching to textured implants than to smooth implants (*p* < 0.001 and *p* = 0.006, respectively) (Figure 2) [83], supporting subclinical infection as a trigger for BIA-ALCL. A longitudinal analysis of 104 cases of BIA-ALCL showed that there is also an increasing incidence of this disease in association with increasing degrees of texture and increasing surface area of the implanted devices [84]. A study of 57 implants removed due to capsular contracture showed that all of the implants had biofilms [85]: lymphocyte counts were correlated with bacterial count and there were significantly more T-cells than B-cells (*p* < 0.001). There was also a significant linear correlation between the numbers of T and B-cells and the numbers of bacteria detected (*p* < 0.001), suggesting lymphocyte activation [85]. The outer envelope of Gram-negative bacteria contains lipopolysaccharides, which have been shown to induce the production of proinflammatory cytokines by multiple cell types (e.g., macrophages and dendritic cells) [86,87,88,89]. Although unproven, this may contribute to the chronic inflammation observed in BIA-ALCL [45]. A full experimental animal model of biofilm causing BIA-ALCL is lacking.

Surgical technique may be a critical factor in preventing BIA-ALCL, similar to capsular contracture prevention [90]. In one study involving over 42,000 macrotextured implants (mean follow-up of 11.7 years for the Biocell implants and 8.0 years for the polyurethane-covered implants), surgeons had consistently used a set of defined surgical techniques (i.e., 14-point plan) designed to mitigate the bacterial load during breast implant procedures [91]. The 14-point plan includes components such as steps to minimize skin contamination, use of IV antibiotics at time of anesthetic induction, use of nipple shields, placing implant in dual plane pocket, and pocket irrigation with antiseptic solution. The expected number of BIA-ALCL cases based on accepted risk norms was 14, yet the actual number of cases was zero. Additional studies to confirm these results and further investigate the associations between surgical technique, bacterial mitigation methods, and the development of BIA-ALCL are warranted.

Although the presence of a given bacterium may be opportunistic rather than causative [92], and interactions between local microbiomes, inflammation, and immunity may be complex [93,94,95], bacteria have been identified as causative agents in other malignancies. For example, *Helicobacter pylori* is known to be a causative factor in gastric B-cell lymphoma [96] and gastric cancer [97] through a mechanism involving chronic infection and inflammation. *Coxiella burnetii*, the causative agent of Q-fever, has been associated with an increased risk of B-cell lymphomas (standardized incidence rate, 25.4) through a mechanism involving immune suppression [98]. In pcALCL, a highly inflammatory toxin released from staphylococcus bacteria serves as a superantigen, that is, an immunostimulatory enterotoxin molecule produced by bacteria stimulates T-cells of a specific β chain variable (Vβ) family regardless of TCR specificity [99]. Superantigenic stimulation of T-cells is selective for cells bearing specific Vβ gene segments of the TCR [100]; however, to date, functional TCRs have not been found in cases of BIA-ALCL [18,28,29], and no precedence exists for Gram-negative bacteria or endotoxins leading to T-cell lymphomas.

A hypothesis involving the role of bacteria in the etiology of BIA-ALCL has been proposed (Figure 3) [62,101]. In this model, colonization by bacteria of textured implants having a high surface area produces a biofilm, which, when the bacterial load exceeds a certain threshold value, leads to chronic antigen stimulation in genetically susceptible individuals. Eventually transformation occurs, leading to the emergence and proliferation of monoclonal CD30-positive ALK-negative T-cells and the development of BIA-ALCL. Dysregulation of the JAK1/STAT3 pathway in affected cells is likely involved, and the observed process is a slow one, typically requiring 8 to 12 years between implantation and BIA-ALCL diagnosis [62,102].

The theory of bacteria in the development of BIA-ALCL has several implications. First, the correlation between the extent of surface texturing, bacterial growth, and BIA-ALCL incidence suggests that a standardized system for classifying breast implant outer shell texture should be adopted to facilitate further research. Such a system, which groups implant surface area measurements into high, intermediate, low, and minimal, allowing for more accurate comparison than existing terms (macrotexture, microtexture, etc.), has been proposed [103].

Although some data support a bacterial role in the pathogenesis of BIA-ALCL, it has not achieved universal acceptance and should not be presented in isolation from other influences, such as possible genetic factors. Newly proposed triggers not discussed here, such as viruses [104,105], may also contribute to the development of BIA-ALCL and warrant further investigation. Ultimately, collaborative research on this uncommon disease will enable the medical community to continue to advance understanding and knowledge with the goal of optimizing patient care.

## 5. Conclusions

The development of BIA-ALCL involves the complex interplay of multiple internal and external factors. Although much regarding its etiology remains to be elucidated, at present, the most developed line of research supports the bacterial hypothesis by which biofilm elicits an inflammatory immune response, eventually resulting in malignant transformation of cells. Among the many unknowns are the specific antigen or superantigen involved, detailed pathways to transformation, particular genetic factors associated with an increased risk of BIA-ALCL, and possible change in classification to lymphoproliferative disorder or possible differences in etiology between effusion-limited BIA-ALCL and the poorer prognosis infiltrative and metastatic BIA-ALCL. Development of an experimental animal model and ongoing translational and clinical research will further elucidate the pathobiology and guide the avoidance and treatment of this disease.

## Figures and Tables

**Figure 1 cancers-12-03861-f001:**
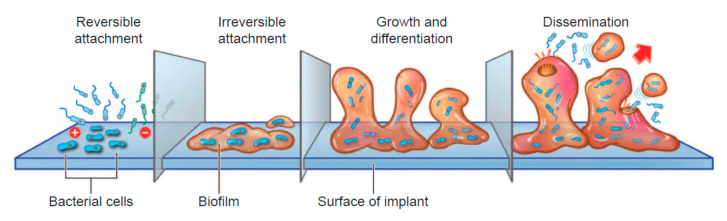
Stages of biofilm growth [78].

**Figure 2 cancers-12-03861-f002:**
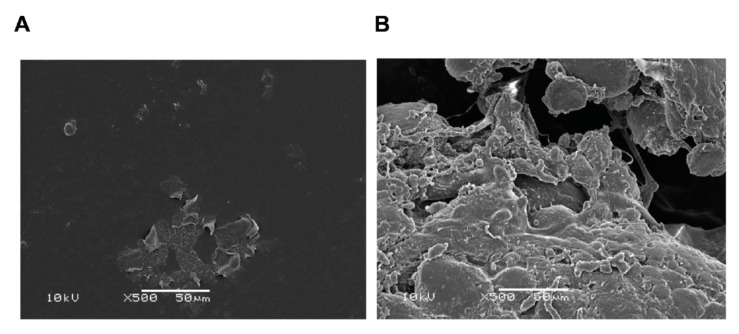
Scanning electron micrographs of biofilm on (**A**) smooth and (**B**) textured implants in female pigs and inoculated with a human strain of *Staphylococcus epidermidis*. Results were confirmed by quantitative polymerase chain reaction [83].

**Figure 3 cancers-12-03861-f003:**
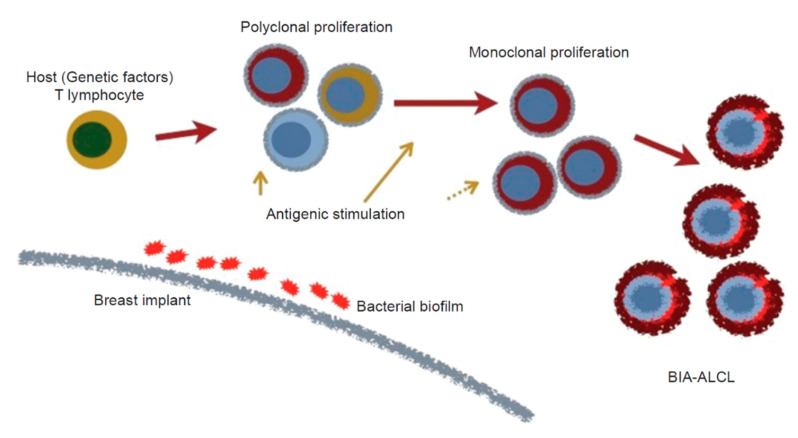
Proposed hypothesis for the genesis of BIA-ALCL [101].

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
