# Peer review of "Etiology of Breast Implant-Associated Anaplastic Large Cell Lymphoma (BIA-ALCL): Current Directions in Research"

_cancers, 2020, doi:10.3390/cancers12123861_

Round 1
Reviewer 1 Report
The authors have totally answered my comments.
The review is excellent and ready for publication.
Reviewer 2 Report
I think the authors well described characteristics and etiology of BIA-ALCL from current papers. I have no more comment.
Reviewer 3 Report
The authors have improved the quality of the manuscript and followed the suggestions of the referees, The paper is suitable to be published.This manuscript is a resubmission of an earlier submission. The following is a list of the peer review reports and author responses from that submission.
Round 1
Reviewer 1 Report
The authors wrote here a useful review dedicated first to summarize the current knowledge on the immunological features of Breast-Implant Associated-Anaplastic Large Cell Lymphoma, and second to discuss multiple internal and external factors, which have been proposed to be involved in the etiology of this uncommon disease. Furthermore, the authors, who are experts in the field, highlight what could be done to better understand the pathology, in an attempt to allow its prevention or to improve its clinical management.
The review is highly interesting and clearly written. I recommend its publication with only minor revisions, listed below:
-it is not clear to me if a “capsular contracture” is or is not a consistent step before the development of BIA-ALCL. A scheme showing the spectrum of BIA-ALCL presentation and the potential evolution of the disease could help.
-two questions came to me while reading this review and if their answers are known, maybe they could be included in the review: 1) an 8 to 12-year latency between the breast implant and the development of BIA-ALCL has been classically observed. Is it known if the age and hormonal status of patients could influence the development of the disease? 2) In the bacteria/biofilm hypothesis, is it known if particular bacteria (from the surgery, or from the breast tissue) consistently lead to indolent form or more complicated forms of the disease?
-to facilitate the understanding to a larger audience, I would suggest to add a scheme illustrating a textured implant, annotated with the words: capsule, capsular tissue, implant shell... Also, brief definitions of the terms: “late seroma”, “capsule-restricted BIA-ALCL”, “effusion-limited BIA-ALCL” and possibly more explanations of the “Baker grade” and of the “14-point plan” would be appreciated.
Author Response
1. The authors wrote here a useful review dedicated first to summarize the current knowledge on the immunological features of Breast-Implant Associated-Anaplastic Large Cell Lymphoma, and second to discuss multiple internal and external factors, which have been proposed to be involved in the etiology of this uncommon disease. Furthermore, the authors, who are experts in the field, highlight what could be done to better understand the pathology, in an attempt to allow its prevention or to improve its clinical management. The review is highly interesting and clearly written. I recommend its publication with only minor revisions, listed below: |
We thank Reviewer #1 for these constructive comments and have addressed each one below.
|
2. -it is not clear to me if a “capsular contracture” is or is not a consistent step before the development of BIA-ALCL. A scheme showing the spectrum of BIA-ALCL presentation and the potential evolution of the disease could help. |
Although capsular contracture may be present in some women with BIA-ALCL, it is not the predominate presentation. In one study reporting on 55 cases of BIA-ALCL, 3 women presented with capsular contracture in combination with seroma or a mass while most (42 women) presented with seroma alone (Loch-Wilkinson et al. Plast Reconstr Surg. 2017;140(4):645). We added text to the bacteria/biofilms section that makes this point.
The spectrum of BIA-ALCL presentation and disease evolution are currently the subject of much research. Due to the complexity of studying this subject matter, the information regarding these topics is continually changing. As such, we did not add a figure; however, Figure 3 in the manuscript presents a plausible hypothesis for the genesis of BIA-ALCL.
|
3. -two questions came to me while reading this review and if their answers are known, maybe they could be included in the review: 1) an 8 to 12-year latency between the breast implant and the development of BIA-ALCL has been classically observed. Is it known if the age and hormonal status of patients could influence the development of the disease?
2) In the bacteria/biofilm hypothesis, is it known if particular bacteria (from the surgery, or from the breast tissue) consistently lead to indolent form or more complicated forms of the disease? |
1) There does not appear to be a relationship between age/hormonal status and development of BIA-ALCL. In one study, age was not a predictor for positive finding on MRI (Oliveira et al. Aesthetic Plast Surg. 2020 Sep 21. Epub ahead of print). In another study, BIA-ALCL onset occurred at a later age in breast reconstruction patients compared with breast augmentation patients (Leberfinger et al. JAMA Surg. 2017;152:1161.). This was thought to be due to the younger age at which women typically have breast augmentation procedures. This information has been added to the Introduction.
2) While specific bacteria have been isolated in BIA-ALCL, there is not yet enough information to distinguish which bacteria may lead to indolent or complicated forms of the disease or if there will be a distinction at all. |
4. -to facilitate the understanding to a larger audience, I would suggest adding a scheme illustrating a textured implant, annotated with the words: capsule, capsular tissue, implant shell... Also, brief definitions of the terms: “late seroma”, “capsule-restricted BIA-ALCL”, “effusion-limited BIA-ALCL” and possibly more explanations of the “Baker grade” and of the “14-point plan” would be appreciated. |
We agree that the suggested figure would be an excellent addition to the manuscript. We added this figure as the graphical abstract, which is prominently displayed on the journal’s web site.
To minimize the number of potentially confusing terms used in the manuscript, we removed the word “late” before seroma in the Abstract. The Abstract was the only place that “late seroma” appeared in the manuscript.
To clarify terms used in the manuscript we made the following revisions: -In the Introduction, we revised “For the majority of patients with BIA-ALCL restricted to the capsule, optimal management consists of timely diagnosis and surgical excision of disease, implants and surrounding capsule” to “For the majority of patients with BIA-ALCL restricted to the fibrous capsule surrounding the breast implant (ie, effusion-limited), optimal management consists of timely diagnosis and surgical excision of implants and capsule with negative margins.”
- In the Bacteria/Biofilms section we further defined Baker grade I and grade IV. - The 14-point plan has been described in more detail in the Bacteria/Biofilms section. |
Reviewer 2 Report
This paper reviewed recent papers of BIA-ALCL, which is important lymphoma category. Authors well described characteristics and etiology of BIA-ALCL from current papers. The onset of the lymphoma still has been unclear, but I guess this review article will help to consider a mechanism of BIA-ALCL development.
Author Response
1. This paper reviewed recent papers of BIA-ALCL, which is important lymphoma category. Authors well described characteristics and etiology of BIA-ALCL from current papers. The onset of the lymphoma still has been unclear, but I guess this review article will help to consider a mechanism of BIA-ALCL development. |
We thank Reviewer #2 for this comment. |
Reviewer 3 Report
This is a very interesting expert review about the etiology and pathogenesis of breast implant associated anaplastic large cell lymphoma (BIA-ALCL).
The actual literature is critically discussed and current research is well documented. Helpful are therefor hints and suggestions for further research in this field.
Despite the fact, that the presentation is not a systematic review based upon a standardized searching technique of the literature, but rather based on literature reflected experts opinions and own research intentions, the paper may be helpful and important for many clinicians and researchers working in this field.
The paper is well written and understandable also for non-immunologists, a fact that may be caused by the interdisciplinary formation of the contributing experts.
Author Response
1. This is a very interesting expert review about the etiology and pathogenesis of breast implant associated anaplastic large cell lymphoma (BIA-ALCL). 2. The actual literature is critically discussed and current research is well documented. Helpful are therefore hints and suggestions for further research in this field. 3. Despite the fact, that the presentation is not a systematic review based upon a standardized searching technique of the literature, but rather based on literature reflected experts opinions and own research intentions, the paper may be helpful and important for many clinicians and researchers working in this field. 4. The paper is well written and understandable also for non-immunologists, a fact that may be caused by the interdisciplinary formation of the contributing experts. |
We thank Reviewer #3 for these comments. |